# Systematic bias in evaluating chemical transport models with maximum daily 8-hour average (MDA8) surface ozone for air quality applications: a case study with GEOS-Chem v9.02

Katherine R. Travis[1], Daniel J. Jacob[2,3]

[1]Department of Civil and Environmental Engineering, Massachusetts Institute of Technology, Cambridge, MA, USA
[2]School of Engineering and Applied Sciences, Harvard University, Cambridge, MA, USA
[2]Department of Earth and Planetary Sciences, Harvard University, Cambridge, MA, USA

*Correspondence to*: K. R. Travis (ktravis@mit.edu)

**Abstract.** Chemical transport models frequently evaluate their simulation of surface ozone with observations of the maximum daily 8-hour average (MDA8) concentration, which is the standard air quality policy metric. This requires successful simulation of the surface ozone diurnal cycle including nighttime depletion, but models often have difficulty simulating this diurnal cycle for a number of reasons including (1) vertical grid structure in the surface layer, (2) timing of changes in mixed layer dynamics and ozone deposition velocity across the day-night transition, (3) poor representation of nighttime stratification, (4) uncertainties in ozone nighttime deposition. We analyze the problem with the GEOS-Chem model, taking as representative case study the Southeast US during the NASA SEAC⁴RS aircraft campaign in August-September 2013. The model is unbiased relative to the daytime mixed layer aircraft observations but has a mean +8 ppb bias at its lowest level (65 m) relative to MDA8 surface ozone observations. The bias can be corrected to +5 ppb by implicit sampling of the model at the 10 m altitude of the surface observations. The model does not capture frequent observed occurrences of <20 ppb MDA8 surface ozone on rainy days, possibly because of unaccounted enhancement of ozone deposition to wet surfaces. Restricting the surface ozone evaluation to dry days still shows inconsistencies with MDA8 ozone because of model errors in the ozone diurnal cycle. Restricting the evaluation to afternoon ozone completely removes the bias. We conclude that better representation of diurnal variations in mixed layer dynamics and ozone deposition velocities is needed in models to properly describe the diurnal cycle of ozone.

## 1 Introduction

Ground-level ozone is harmful to human health and vegetation. It is produced when volatile organic compounds (VOCs) and carbon monoxide (CO) are photochemically oxidized in the presence of nitrogen oxide radicals ($NO_x \equiv NO+NO_2$). Ozone air quality standards in different countries are generally formulated using the maximum daily 8-hour average concentration (MDA8) as a metric. In the US, the current ozone National Ambient Air Quality Standard (NAAQS) set by the Environmental Protection Agency (EPA) is 70 ppb as the fourth-highest MDA8 concentration per year, averaged over three years (EPA,

2015). Exceedances of the standard generally occur during daytime due to photochemical production and to entrainment of elevated ozone from aloft (Kleinman, et al., 1994). Ozone is depleted at night due to deposition and chemical loss in a shallow surface layer capped by a stratified atmosphere.

Air quality agencies rely on chemical transport models (CTMs) to identify the most effective emission reduction strategies for ozone pollution. CTMs predict surface ozone concentrations on the basis of $NO_x$, VOC, and CO emissions, accounting for chemistry and meteorological conditions. CTMs tend to overestimate surface ozone, particularly in the Southeast United States (Fiore et al., 2009; Makar et al., 2017). Some of this overestimate is likely due to bias in the $NO_x$ emission inventories (Anderson et al., 2014; Travis et al. 2016), but the choice of comparison metric could also play a role. MDA8 ozone is
commonly used as the metric for evaluating models with observations and making predictions relevant to air quality standards (Fiore et al., 2009; Mueller and Mallard, 2011; Emery et al., 2012; Lin et al., 2012; Rieder et al., 2015). Use of this metric implicitly requires successful simulation of the diurnal cycle in surface ozone but models are often too high at night, apparently because they cannot resolve the local stratification and associated depletion from surface deposition. This is a problem not only in global models with coarse vertical resolution (Lin and McElroy, 2010; Schnell et al., 2015; Strode et al., 2015) but also
in regional air quality models (Herwehe et al., 2011; Solazzo et al., 2012; Solazzo and Galmarini, 2016). A recent evaluation of the CMAQ regional model shows little bias in the diurnal cycle averaged over all monitoring sites in the contiguous US (Appel et al., 2017) but such averaging may smooth the diurnal cycle across different regions (Bowdalo et al., 2016) and across urban, rural, and background sites.

Here we evaluate the use of the MDA8 ozone metric in the GEOS-Chem CTM, a global model frequently used in studies of regional ozone air quality and evaluated for this purpose with MDA8 ozone (Racherla and Adams, 2008; Lam et al., 2011; Zhang et al., 2011; Zoogman et al., 2011; Emery et al., 2012; Zhang et al., 2014). We focus on the Southeast US in summer, where extensive model evaluation with observations of ozone and its precursors was done as part of the NASA SEAC[4]RS aircraft campaign (Travis et al., 2016). After correcting for bias in $NO_x$ emissions, Travis et al. (2016) found that the model
had no significant ozone bias relative to aircraft observations below 1 km altitude but still overestimated MDA8 surface ozone by +6 ppb on average. As we show here, this may largely be explained by the inability of the model to represent nighttime ozone depletion from the shallow surface layer. The ultimate solution of this problem will require improved representation of boundary layer physics, but we propose in the meantime some simple corrective measures.

## 2 Bias in simulation of MDA8 surface ozone

We use the GEOS-Chem simulation previously applied by Travis et al. (2016) to interpret observations from the SEAC[4]RS aircraft campaign in August-September 2013 (Toon et al., 2016). The simulation is based on GEOS-Chem version 9.02 with detailed oxidant-aerosol chemistry ([www.geos-chem.org](www.geos-chem.org)) and is driven by assimilated meteorological data from the Goddard

Earth Observing System – Forward Processing (GEOS-FP) product of the NASA Global Modeling and Assimilation Office (GMAO) using the GEOS-5.11.0 general circulation model (Molod et al., 2012). The GEOS-FP data have a native horizontal resolution of 0.25° latitude by 0.3125° longitude, with 72 levels in the vertical extending up to the mesosphere on a hybrid sigma-pressure grid and a temporal resolution of one hour for surface variables and mixing depths. The lowest levels are centered at 65 m, 130 m, 200 m, and 270 m above ground level (AGL). Boundary layer turbulence follows the clear-sky non-local parameterization from Holtslag and Boville (1993), as implemented in GEOS-Chem by Lin and McElroy (2010). Dry deposition of ozone follows a standard resistance-in series scheme (Wesely, 1989; Wang et al., 1998) where the surface resistance depends on leaf area and stomatal opening (itself dependent on temperature and solar radiation). The native 0.25°×0.3125° resolution is used in GEOS-Chem over North America and adjacent oceans (130° - 60° W, 9.75° - 60° N), with boundary conditions from a global simulation with 4°×5° horizontal resolution. Detailed evaluations of GEOS-Chem with observations over the Southeast US for the SEAC[4]RS period are presented in other papers (Kim et al., 2015; Fisher et al., 2016; Marais et al., 2016; Yu et al., 2016; Zhu et al., 2016; Miller et al., 2017). Specific evaluation for ozone and related species is presented in Travis et al. (2016).

Travis et al. (2016) found that despite successful simulation of ozone and its precursors in the SEAC[4]RS aircraft data below 1 km altitude, MDA8 surface ozone was biased high in the model by +6 ppb on average. Fig. 1 (left panel) shows the frequency distributions of ozone concentrations measured by the aircraft in the mixed layer below 1 km during afternoon hours (12-17 local solar time or LT) and simulated by the model along the flight tracks and at the flight times. The data have been filtered for biomass burning ($CH_3CN > 200$ ppt) and urban plumes ($NO_2 > 4$ ppb), which the model would not be expected to capture. The bias between the model and observations is small (+2 ppb) and within statistical uncertainty ($p$=0.07). The center panel of Fig. 1 shows the observed and simulated frequency distributions of daily MDA8 surface ozone in August-September 2013 at the thirteen rural CASTNET sites in the Southeast US (EPA, 2018), with the model sampled at the lowest model grid level ($z_m$ = 65 m AGL). The Southeast US region is a relatively coherent region for surface ozone, with different sites showing similar behaviors (Bowdalo et al., 2016). The model is biased high by +8 ppb on average and this is highly significant ($p < 0.01$). The bias differs slightly from the +6 ppb in Travis et al. (2016) who showed a comparison for June-August versus August-September here. Comparison of the mean ozone concentrations in the mixed layer (aircraft afternoon data below 1 km) and at the surface (MDA8) indicates a vertical difference of 9 ppb in the observations but only 3 ppb in GEOS-Chem.

**3 Correcting for surface layer gradients**

A first problem in comparing the model to the CASTNET surface air observations is the mismatch between the lowest model level midpoint ($z_m$ = 65 m AGL) and the level at which the observations are made ($z_l \approx 10$ m AGL). This can be corrected easily because the model implicitly simulates an ozone concentration at $z_l$ through the aerodynamic resistance $R_a(z_l, z_m)$ to turbulent vertical transfer in the resistance-in-series parameterization of dry deposition (Brasseur and Jacob, 2017). The model

calculates a local ozone deposition velocity $v_d(z_m)$ at altitude $z_m$ assuming uniform vertical flux down to the surface. We can then infer the implicit model ozone concentration $C(z_1)$ at 10 m from the explicit concentration $C(z_m)$ at 65 m (Zhang et al., 2012):

$$C(z_1) = (1 - R_a(z_1, z_m)v_d(z_m))C(z_m), \tag{1}$$

$R_a(z_1, z_m)$ is calculated in GEOS-Chem by similarity with momentum for a neutral atmosphere (friction velocity $u^*$) including a heat-based stability correction $\phi_h(z/L)$, where $L$ is the Monin-Obukhov length:

$$R_a = \int_{z_1}^{z_m} \frac{\phi_h(z/L)}{ku^*z} dz, \tag{2}$$

Here $k = 0.4$ is the von Karman constant. Equations 3(a-c) describe $\phi_h$, from Dyer (1974) for unstable and moderately stable conditions ($z/L < 1$) and from Holtslag et al. (1990) for stable conditions ($z/L > 1$):

$$\phi_h = 5 + z/L, \qquad z/L > 1 \tag{3a}$$

$$\phi_h = 1 + 5\,z/L, \qquad 0 < z/L < 1 \tag{3b}$$

$$\phi_h = (1 - 16\,z/L)^{-1/2}, \qquad z/L < 0 \tag{3c}$$

The model deposition velocity $v_d(z_m)$ over the Southeast US during SEAC[4]RS averages $0.7 \pm 0.3$ cm s$^{-1}$ in daytime, consistent with observations (Travis et al., 2016). Applying the transfer function from equation (1) at the CASTNET sites we find a mean

MDA8 model concentration at 10 m altitude of $45 \pm 8$ ppb, as compared to $48 \pm 9$ ppb at 65 m. Correcting the model to 10 m altitude thus decreases the model bias relative to observations by 3 ppb, but a bias of +5 ppb remains. Model MDA8 ozone at 65 m has ten exceedances of the 70 ppb NAAQS for the CASTNET data in Fig. 1, as compared to one exceedance in the observations, and sampling the model at 10 m decreases the number of model exceedances to four.

## 4 Segregating rainy conditions

The most severe bias in comparing the model MDA8 ozone to the CASTNET observations in Fig. 1 is for the low tail of the distribution (ozone below 25 ppb). 7 % of observed MDA8 ozone values are below 25 ppb ($n = 49$) but there is only one value below 25 ppb in the model at either 65 or 10 m. This low-tail model bias has been found before (Fiore et al., 2002; McDonald-Buller et al., 2011) and attributed to inflow of low-ozone tropical air from the Gulf of Mexico. However, our model simulation is unbiased over the Gulf of Mexico relative to the SEAC[4]RS aircraft observations (Travis et al., 2016). In addition, the

occurrence of low values of observed MDA8 ozone is distributed across the CASTNET sites in the Southeast and is not related to distance from the Gulf.

We find instead that the low MDA8 ozone values in the CASTNET observations are associated with rainy conditions and that rain has less of an effect on ozone in the model. Figure 2 segregates the frequency distribution of MDA8 ozone at CASTNET

sites between rainy days and dry days. Rainy days are defined by 24-h total rainfall exceeding 6 mm and dry days by 24-h total rainfall less than 1 mm. Rainy and dry days are diagnosed in the observations with the high-resolution data from the Parameter-elevation Regressions on Independent Slopes Model (PRISM) climate group (PRISM, 2016) regridded to the model resolution of 0.25º × 0.3125º. Rainy and dry days in the model are diagnosed from the GEOS-FP data, and do not necessarily coincide with rainy and dry days in the observations; our purpose here is to compare how rain affects ozone in the observations and in the model. 15% of observation days and 10% of model days are rainy. Observed ozone on rainy days averages 9 ppb lower than on dry days (33 vs 42 ppb). Model ozone is on rainy days averages only 5 ppb lower than on dry days (41 vs 46 ppb). Rainy conditions can cause MDA8 ozone to drop below 20 ppb in the observations but not in the model. Depletion of surface ozone under rainy conditions is not due to wet scavenging, considering the low solubility of ozone in water. It may instead reflect increased atmospheric stability from surface evaporative cooling, combined with increased ozone dry deposition on wet surfaces (Finkelstein et al., 2000; Altimir and Kolari, 2006; Potier et al., 2017; Clifton et al., 2019) that is not considered in our standard surface resistance model for dry deposition. Excluding all rainy days in the comparison of model to observations for MDA8 ozone decreases the model mean bias modestly from +5 ppb to +4 ppb, but more importantly it excludes the low tail of the observed distribution that the model cannot capture.

## 5 Accounting for diurnal bias

Yet another factor in the model overestimate of MDA8 surface ozone is the poor simulation of the diurnal cycle. Figure 3 shows the average ozone diurnal cycle for dry days in the model and in the observations at the CASTNET sites of Fig. 1. The observations show maximum values in the afternoon (14-16 LT) and a gradual decrease at night to a mean minimum value of 17 ppb at 7 LT. The nighttime depletion cannot be due to chemical titration by anthropogenic NO emissions since the selected CASTNET sites are rural and not located near major roadways or industrial sources. It must instead be due to deposition, including possible titration by short-lived biogenic VOCs (Goldstein et al., 2004; Ruuskanen et al., 2011; Rossabi et al., 2018) under stratified surface layer conditions. The model diurnal cycle at 65 m altitude (lowest model level) has the correct phase but the amplitude is much too weak. Correcting the model to 10 m altitude (thus accounting for the vertical gradient within the lowest model level, including for stable conditions as given by equations (1)+(2)+(3c)) increases the amplitude but nighttime depletion is still insufficient. The difference between 65 and 10 m grows rapidly in late afternoon between 16 and 18 LT as the atmosphere becomes stable ($L > 0$) but ozone deposition is still fast because of open stomata. After the stomata close at night the gradient weakens. We find negligible difference in the model diurnal cycle shown in Fig. 3 between August and September. Lack of diurnal cycle in modeled anthropogenic emissions has been suggested as a cause of the general underestimate among models of the summertime diurnal amplitude of ozone concentrations (Schnell et al., 2015), but the emissions used here have hourly resolution based on the National Emission Inventory of the US Environmental Protection Agency. We conclude that the insufficient nighttime depletion in the model must be due to insufficient vertical stratification of the surface layer, together with possible underestimate of nighttime deposition (Musselman and Minnick, 2000;

Lombardozzi et al, 2017). The large ozone bias in the evening hours may reflect small errors in the correlated timing between day-night transition to stable conditions and stomata closure.

The poor model representation of the ozone diurnal cycle implies that the model may err in the diurnal timing of MDA8 ozone. Fig. 4 shows the frequency distribution of the beginning of the 8-hour interval for MDA8 ozone at the CASTNET sites on dry days, comparing the observations and the model. The frequency distribution in the observations peaks sharply at 11 LT (MDA8 window of 11-18 LT), consistent with the mean diurnal cycle of Fig. 3. The model sampled at 65 m also has a maximum probability of MDA8 ozone starting at 11 LT, but also a secondary maximum at 19 LT that is absent from the observations. The latter conditions occur in the model when the atmosphere becomes stable already at 16 LT, decoupling 65 m from the surface and the associated deposition. Under these conditions the model concentration at 65 m remains high in the evening and at night. Correcting the model calculation of MDA8 to use the 10-m ozone largely removes this secondary maximum (Fig. 4) but shifts the peak occurrence of MDA8 ahead by two hours (starting at 9 LT) because of the exaggerated model drop at 17 LT when the model atmosphere becomes stable but ozone stomatal deposition is still active (Fig. 3). The transition from a convective mixed layer to stable nighttime conditions is difficult for models to capture and is an active area of research (Lothon et al., 2014). The correlated timing with stomatal closure further complicates the simulation of the day-night transition in surface ozone.

Model error in the simulation of the ozone diurnal cycle due to insufficient nighttime depletion thus induces a representation error when comparing to MDA8 observations, as the MDA8 periods in the model do not correspond to the same times of day as in the observations. This causes positive bias in the comparison. Another approach in model evaluation is to focus instead on afternoon conditions, recognizing that the model is inadequate to simulate ozone depletion in the shallow surface layer at night (e.g., Fiore et al., 2002). The right panel of Fig. 1 compares simulated and observed frequency distributions of surface ozone at the CASTNET sites at 12-17 LT on dry days, sampling the model at 10 m altitude. The +8 ppb bias in the original model comparison (center panel) is reduced to only +1 ppb. Focusing evaluation on afternoon hours can be adequate for understanding general properties of the model ozone budget, such as the response to changes in $NO_x$ emissions (Strode et al., 2015), because the stratified surface layer represents only a small volume of atmosphere. However, the problem of simulating the policy-relevant MDA8 surface ozone remains.

## 6 Implications

We identified three modeling problems biasing the comparison to observed maximum daily 8-h average (MDA8) ozone for air quality applications: (1) vertical mismatch between the lowest model level and the altitude of the observations, (2) insufficient vertical stratification and/or ozone loss (e.g., non-stomatal dry deposition pathways) under rainy conditions or at night, and (3) inadequate representation of the day-night transition to stable conditions leading to error in timing of the 8-hour

MDA8 window. Problem (1) can be solved by using the parameterization of surface layer turbulence implicit in the model simulation of dry deposition, although the parameterization may underestimate the vertical gradient under stable conditions. Finer vertical grid resolution of the surface layer in the parent GEOS-5 dynamical model for GEOS-Chem could improve the representation of the gradient. Problems (2) and (3) suggest the need for more research in the dynamics of stable boundary layers and in the deposition of ozone to wet surfaces and at night. Fine temporal consistency in the modeling of mixed layer dynamics and chemical deposition fluxes across the day-night transition is also important. Focusing model evaluation on dry afternoon conditions circumvents these problems and is mostly adequate for general testing of the model ozone chemistry. Further model evaluation with MDA8 ozone for air quality applications should be contingent on proper representation of the ozone diurnal cycle.

## 7 Data availability

PRISM temperature and precipitation data can be downloaded at http://www.prism.oregonstate.edu/historical/. CASTNET observations are available here: https://www.epa.gov/castnet. SEAC$^4$RS aircraft observations are available here: https://www-air.larc.nasa.gov/cgi-bin/ArcView/seac4rs. The model code is available here: https://github.com/ktravis213/SEAC4RS_V10. The hourly model output is available here:

https://www.dropbox.com/s/hzoy19do4mw41rn/Travis_GMD_2019_GEOSChem_timeseries.zip?dl=0.

*Author Contributions*

KRT and DJJ designed this study and prepared the manuscript.  KRT performed the simulations and analyses.

*Acknowledgements*

Thank you to Thomas Ryerson, Ilana Pollack, Jeff Peischl for the use of their ozone data from the NOAA $NO_yO_3$ instrument. We acknowledge Christoph Keller for his useful comments on calculating 10-m model ozone and Melissa Puchalski for her help with using hourly CASTNET data. This research was supported by the NASA Atmospheric Composition Modeling and Analysis Program.

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

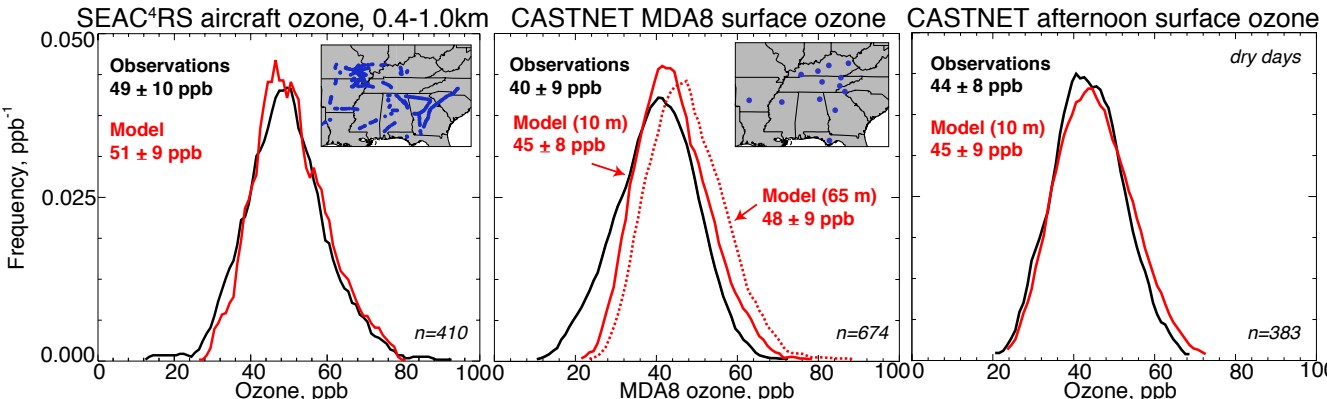

**Figure 1:** Frequency distributions of ozone concentrations in the Southeast US (94.5-80 W, 29.5-38 N) in August-September 2013, sampled at the blue locations in the maps inset. Observations are compared to GEOS-Chem model values sampled at the same locations and times. Means and standard deviations are given inset. The left panel shows afternoon (12-17 local solar time) mixed layer values from the SEAC⁴RS DC8 aircraft at 0.4-1.0 km altitude. Ozone measurements are from the NOAA NOyO3 four-channel chemiluminescence (CL) instrument (Ryerson et al., 1998) The center panel shows MDA8 surface ozone at the CASTNET network of 13 rural sites, compared to the model sampled at the lowest model gridpoint 65 m above ground (dashed line)and the inferred model value at 10 m (solid line) as described in the text. The right panel shows afternoon ozone at the CASTNET sites excluding days with rain in either the model or the observations.

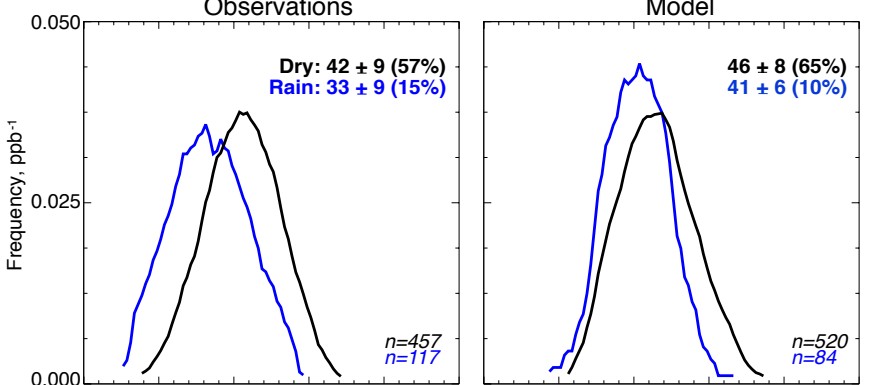

**Figure 2:** Frequency distributions of MDA8 ozone at CASTNET sites in the Southeast US in August-September 2013, segregating rainy and dry days as described in the text. The model is sampled at 10 m altitude to match observations, as described in Section 3. Mean ozone and its standard deviation are given inset, with the percentages of dry and rainy days in parentheses. The percentages do not add to 100 % because of additional contribution from marginal days where rainfall is between 1 and 6 mm.

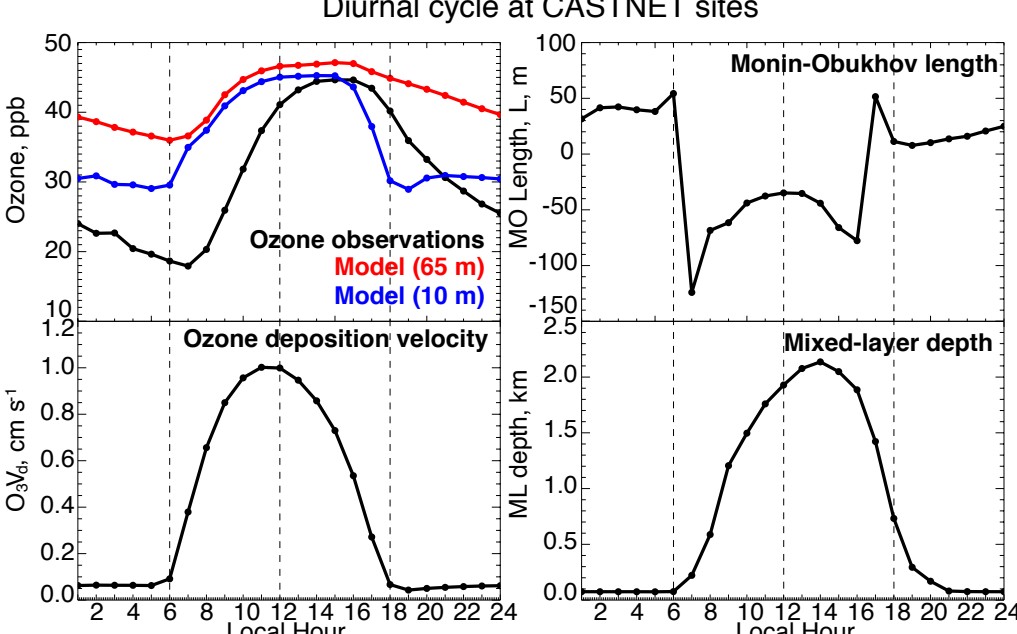

**Figure 3:** Mean diurnal cycle of ozone and related surface variables at the 13 Southeast US CASTNET sites in Fig. 1 for August-September 2013. Ozone observations in the top left panel are compared to GEOS-Chem values sampled at 65 m altitude (lowest model level) and at 10 m altitude (where the observations are sampled). Other panels show the mean 10-m ozone deposition velocity in GEOS-Chem, the median Monin-Obukhov length L in the GEOS-FP data used to drive GEOS-Chem, and the mean mixed layer depth in the GEOS-FP data. Days where precipitation exceeds 1 mm in either the model or observations are excluded. Local hour refers to solar time (maximum solar elevation at noon). Vertical dashed lines at 6, 12, and 18 local time are to guide the eye.

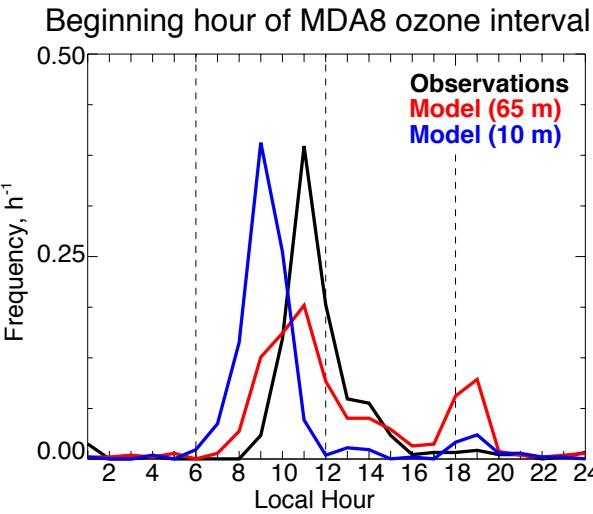

**Figure 4:** Timing of MDA8 ozone at the Southeast US CASTNET sites in August-September 2013. The figure shows the frequency distributions of the beginning hour of the 8-hour period defining the MDA8 ozone value for each day. Only dry days (24-h precipitation less than 1 mm) are included.