# Peer review of "Systematic bias in evaluating chemical transport models with maximum daily 8-hour average (MDA8) surface ozone for air quality applications: a case study with GEOS-Chem v9.02"

_Geoscientific Model Development, 2019_

## Referee Comment (RC1) · Anonymous Referee #1 · 14 Apr 2019

It was a pleasure to read the manuscript by Travis and Jacob and follow their analysis on GEOS-Chem model biases with respect to simulations of ground-level ozone concentrations and their diurnal cycle. The text is well structured and clearly written, the figures are of good quality, and the science is sound and well documented. I therefore suggest publication of this work after minor revision as detailed below.

Abstract: the wording of the abstract could be improved to better emphasize the identification of three problems that are to some extent linked to each other as it is presented in the Discussion. In particular, the interplay between statistics (MDA8 sampling interval) and science, or model deficit (transition from daytime to nighttime boundary layer) could be made more explicit.

p2, l. 29: please mention model top (72 levels from surface to where?)

p3, l.4: remove final semi-colon in citation

p3, l.11 – don't speak about "statistical significance" (see https://www.nature.com/articles/d41586-019-00874-8)

p5, l.25: replace "pdf" by "histogram" as you show discrete hours in figure 4

p5, l.29: I don't understand this argument: According to the model description, 65 m is the center altitude of the lowest model layer. It is this model layer on which dry deposition will act to reduce ozone concentrations – so, how can 65 m be "decoupled from the surface" in the model?

p6, l.12: remove "insignificant" (see above)

p6, l.22 an obvious solution here could appear to increase model vertical resolution near the surface (for example, the ECMWF model has 10 m as its lowest center altitude). This option should probably be mentioned and perhaps briefly discussed.

p7, l.4: the model data (at least a reasonable subset that allows to reproduce the results, for example, time series extracted at the CASTNET locations) must also be made available.

figure 4 - caption: what is shown here is a histogram rather than a pdf since you have discrete hour values
* * *

---

## Short Comment (SC1) · 24 Apr 2019

Dear authors,

In my role as Executive editor of GMD, I would like to bring to your attention our Editorial version 1.1:

http://www.geosci-model-dev.net/8/3487/2015/gmd-8-3487-2015.html

This highlights some requirements of papers published in GMD, which is also available

on the GMD website in the 'Manuscript Types' section:

http://www.geoscientific-model-development.net/submission/manuscript_types.html

In particular, please note that for your paper, the following requirements have not been met in the Discussions paper:

- "The main paper must give the model name and version number (or other unique identifier) in the title."

- "If the model development relates to a single model then the model name and the version number must be included in the title of the paper. If the main intention of an article is to make a general (i.e. model independent) statement about the usefulness of a new development, but the usefulness is shown with the help of one specific model, the model name and version number must be stated in the title. The title could have a form such as, "Title outlining amazing generic advance: a case study with Model XXX (version Y)"."

As GEOS-Chem seems to be the only model used in your study, please add its name and version number to the title of your manuscript, e.g., "Systematic bias in evaluating chemical transport models with maximum daily 8-hour average (MDA8) surface ozone for air quality applications: a case study with GEOS-Chem vX.y"

Yours,

Astrid Kerkweg

---

## Referee Comment (RC2) · Anonymous Referee #2 · 29 Apr 2019

I find the the paper well written and presented, results clearly explained.

Although the scopes are overall valuable, I believe that the paper is too limited in the analysis in its current shape.

The authors focused on the results of one model applied to a rather 'narrow' dataset to derive conclusions that have been known for a while (I would actually say that the PBL transition is THE long standing issue for atmospheric dispersion models at al scales). The authors have the merit of having managed to isolate the portion of data

that serves to clearly illustrates their points (daily cycle and PBL transition), but have not substantiate their conjectures with additional model runs (for instance switching deposition on/off to check the conjectures of section 5) and/or additional observation (longer time periods, data from another region, ...), and/or other models.

My impression is that the paper, as it stands, lacks of robustness and seems more a technical report on GEOS-Chem than a stand-alone scientific publication. I would therefore invite the authors to expand the analysis to other data or sensitivity runs in support of your conclusions.

SPECIFIC COMMENTS Section 4. How does the model perform for precipitation? from my understanding (but I might be wrong) you look at ozone performance conditioned to rain or no rain condition. But you need first to check if the model is 'doing the right (or wrong) thing for the right reason', and thus you should give information to the reader on how the model catches rainy conditions.

How is 'rainy conditions' defined (threshold, number of hours, ...)? how many occurrences are there over the examined periods?

Additional references for the authors to consider: MAkar et al., Nature Communications volume 8, Article number: 15243 (2017);

Dennis et al, ON THE EVALUATION OF REGIONAL-SCALE PHOTOCHEMICAL AIR QUALITY MODELING SYSTEMS

---

## Referee Comment (RC3) · Anonymous Referee #3 · 12 Jun 2019

Review of "Systematic bias in evaluating chemical transport models with maximum daily 8-hour average (MDA8) surface ozone for air quality applications" by Travis and Jacob.

This manuscript documents several contributors to model biases in a commonly used metric for ozone air quality (MDA8), and proposes several short-term and long-term methods for addressing this bias. While this paper focuses exclusively on a single model, biases in MDA8 ozone over the region and season studied (the Southeast US during summer/fall) is a pervasive problem in many current atmospheric chemistry

models. Presumably, this same issue would affect model-observation comparisons in other regions and seasons, as well.

The main conclusions of the study are that the biases arise from: (1) diagnostic mismatch of sampling altitudes between model and observations, (2) failure of model to produce sufficiently low ozone concentrations under rainy conditions, and (3) poor representation in the model of the diurnal cycle in boundary layer mixing and in stomatal conductance. Of these issues, (1) is the most straightforward to address, and the paper suggests an approach to diagnose more accurately the simulated ozone concentration at the altitude of the observations. This is a good recommendation that should be considered for adoption in other air quality modeling studies. The mechanisms responsible for issue (2) are not adequately addressed in the paper. The focus in the paper is on increased vertical stability resulting from evaporative cooling, but an alternative hypothesis of increased (non-stomatal) deposition of ozone under wet conditions is not adequately explored. Finally, the paper proposed to address issue (3) by focusing comparisons on afternoon ozone values rather than MDA8 in the short term, and by improving the representation of boundary layers (and presumably stomatal conductance) in the longer term.

This paper, with sufficient revisions, could provide a useful contribution to the literature and would help to address a long-standing bias of atmospheric chemistry models in simulating surface ozone. Specific comments and suggestions are included below.

1. Introduction

page 2, lines 6-7 – Mention that the use of MDA8 for comparisons between models and observations was intended to remove (some of) the known biases in the simulation of nighttime ozone, as opposed to comparing 24-hour averages.

2. Comparing simulations of mixed layer and MDA8 surface ozone

p.3, l. 9 – Is this different from how local solar time is treated in the observations?

p.3, l.14 – Add "sampled at lowest model grid level (zm=65m AGL)" here.

3. Correcting for surface layer gradients

p.4, l.5 – Not really a "correction." Instead, it is a transfer function from z=65m to z=10m.

4. Segregating rainy conditions

p.4, l. 14-16 – But, this doesn't establish that transport from GoM to SE US is correct in model (e.g., nighttime low-level jet).

p.4, l. 19-23 – How similar are the dates diagnosed as rainy/dry in the model vs obs? That is, how well does the model simulate daily variability of precipitation?

p.4, l.27 – Add "increased" before "vertical stratification."

p.4, l.27-29 – See also Clifton et al. (2017), who say:

Recent field-based evidence suggests that nonstomatal processes include ... aqueous chemical reactions on vegetation and soil [Fowler et al., 2009; Ganzeveld et al., 2015; Fumagalli et al., 2016].

p.4, l. 27-29 and elsewhere – Need more description of GC dry deposition scheme. In particular, how does dry deposition velocity respond to moisture (incl. rainfall, soil moisture, dew on leaves, relative humidity, vapor pressure deficit)? Are there potentially missing processes that could increase ozone deposition velocities under wet conditions?

5. Accounting for diurnal bias

p.5, l.6 – Or other large NOx emission sources?

p.5, l. 8-12 – Couldn't this also result from (excessive) mixing of ozone from throughout the first model grid level down to the surface. The rescaling to 10-m values wouldn't correct for this. Also, how valid are the assumption used in this rescalaing under stable nighttime conditions?

p.5, l. 11-12 – Explain what drive the (diurnal) variations in stomatal conductance in GC.

6. Implications

p.6, l.15 – Add "(e.g., non-stomatal dry deposition pathways)" here.

p.6, l.15-16 – Is the evening bias in models due exclusively to errors in vertical mixing, or could errors in the timing of the shutdown of stomatal conductance also play a role?

p.6, l. 17-18 – Is better near-surface vertical resolution in models needed?

p.6, l.22 – Not discussing predictions elsewhere in paper. Change "predicted with confidence" to "simulated more accurately."

---

## Author Comment (AC1) · 28 Jun 2019

We thank the three reviewers and the editor for their careful reading of the manuscript and their detailed comments. Our responses are shown below in blue, with new text in bold.

**Editor**

"The main paper must give the model name and version number (or other unique identifier) in the title." • "If the model development relates to a single model then the model name and the version number must be included in the title of the paper. If the main intention of an article is to make a general (i.e. model independent) statement about the usefulness of a new development, but the usefulness is shown with the help of one specific model, the model name and version number must be stated in

the title. The title could have a form such as, "Title outlining amazing generic advance: a case study with Model XXX (version Y)"."

As GEOS-Chem seems to be the only model used in your study, please add its name and version number to the title of your manuscript, e.g., "Systematic bias in evaluating chemical transport models with maximum daily 8-hour average (MDA8) surface ozone for air quality applications: a case study with GEOS-Chem vX.y" We added ": a case study with GEOS-Chem v9.02" to the title.

**Anonymous Referee #1**

Abstract: the wording of the abstract could be improved to better emphasize the identification of three problems that are to some extent linked to each other as it is presented in the Discussion. In particular, the interplay between statistics (MDA8 sampling interval) and science, or model deficit (transition from daytime to nighttime boundary layer) could be made more explicit.

We re-wrote the Abstract and particularly added the following text to address the reviewer's comment on p1, line 14 "...often have difficulty simulating this diurnal cycle for a number of reasons including (1) vertical grid structure in the surface layer, (2) timing of changes in mixed layer dynamics and ozone deposition velocity across the day-night transition, (3) poor representation of nighttime stratification, (4) uncertainties in ozone nighttime deposition."

p2, l. 29: please mention model top (72 levels from surface to where?) We added "**extending up to the mesosphere**" on page 3, l. 7.

p3, l.4: remove final semi-colon in citation Removed.

p3, l.11 – don't speak about "statistical significance" (see https://www.nature.com/articles/d41586-019-00874-8) Changed "not statistically significant" to "**within statistical uncertainty**".

p5, 1.25: replace "pdf" by "histogram" as you show discrete hours in figure 4 Replaced all instances of pdf to **frequency distribution** which more accurately describes the presentation of our data.

p5, l.29: I don't understand this argument: According to the model description, 65 m is the center altitude of the lowest model layer. It is this model layer on which dry deposition will act to reduce ozone concentrations – so, how can 65 m be "decoupled from the surface" in the model?

We slightly altered the text on p4, line 6 to clarify this point – "This can be corrected easily because the model implicitly simulates an ozone concentration at  $z_1$  through the aerodynamic resistance  $R_a(z_1, z_m)$  to turbulent vertical transfer in the resistance-in-series parameterization of dry deposition (Brasseur and Jacob, 2017)." We also added the following clarifying text on p6, l.5. "(thus accounting for the vertical gradient within the lowest model level, including for stable conditions as given by equations (1)+(2)+(3c))"

P5, 1.7: remove "insignificant" (see above) Replaced "an insignificant" with "**only**". p6, 1.22 an obvious solution here could appear to increase model vertical resolution near the surface (for example, the ECMWF model has 10 m as its lowest center altitude). This option should probably be mentioned and perhaps briefly discussed.

We have added the following discussion on p7, 1.18– "Finer vertical grid resolution of the surface layer in the parent GEOS-5 dynamical model for GEOS-Chem could improve the representation of the gradient."

p7, 1.4: the model data (at least a reasonable subset that allows to reproduce the results, for example, time series extracted at the CASTNET locations) must also be made available. We added a link to the hourly model output on p7, 1.4. **"The hourly model output is available here:** https://www.dropbox.com/s/hzoy19do4mw41rn/Travis GMD 2019 GEOSChem timeseries.zip?dl=0."

figure 4 - caption: what is shown here is a histogram rather than a pdf since you have discrete hour values Replaced all instances of pdf to **frequency**, since we do not actually have a histogram.

**Anonymous Referee #2**

I find the the paper well written and presented, results clearly explained. Although the scopes are overall valuable, I believe that the paper is too limited in the analysis in its current shape.

The authors focused on the results of one model applied to a rather 'narrow' dataset to derive conclusions that have been known for a while (I would actually say that the PBL transition is THE long standing issue for atmospheric dispersion models at alscales). The authors have the merit of having managed to isolate the portion of data that serves to clearly illustrates their points (daily cycle and PBL transition), but have not substantiate their conjectures with additional model runs (for instance switching deposition on/off to check the conjectures of section 5) and/or additional observation (longer time periods, data from another region, ...), and/or other models. My impression is that the paper, as it stands, lacks of robustness and seems more a technical report on GEOS-Chem than a stand-alone scientific publication. I would therefore invite the authors to expand the analysis to other data or sensitivity runs in support of your conclusions.

We now state specifically in the title that this is a GEOS-Chem case study, and explain in the Introduction how it is prompted by the unique set of observational constraints available over the Southeast US in August-September 2013. At the same time, we emphasize how the results have general implications for other models and conditions.

**SPECIFIC COMMENTS Section 4.**

How does the model perform for precipitation? From my understanding (but I might be wrong) you look at ozone performance conditioned to rain or no rain condition. But you need first to check if the model is 'doing the right (or wrong) thing for the right reason', and thus you should give information to the reader on how the model catches rainy conditions. How is 'rainy conditions' defined (threshold, number of hours, ...)? how many occurrences are there over the examined periods?

The definition of "rainy conditions" is given in Section 4. "Rainy days are defined by 24-h total rainfall exceeding 6 mm and dry days by 24-h total rainfall less than 1 mm.". We add the following clarifying text on p5,1.18. "**15% of observation days and 10% of model days are rainy.**"

Additional references for the authors to consider: MAkar et al., Nature Communications volume 8, Article number: 15243 (2017);

We cite Makar et al, 2017 now on page 2, line 11.

Dennis et al, ON THE EVALUATION OF REGIONAL-SCALE PHOTOCHEMICAL AIR QUALITY MODELING SYSTEMS This appears to be unpublished work.

**Anonymous Referee #3**

This manuscript documents several contributors to model biases in a commonly used metric for ozone air quality (MDA8), and proposes several short-term and long-term methods for addressing this bias. While this paper focuses exclusively on a single model, biases in MDA8 ozone over the region and season studied (the Southeast US during

summer/fall) is a pervasive problem in many current atmospheric chemistry models. Presumably, this same issue would affect model-observation comparisons in other regions and seasons, as well.

The main conclusions of the study are that the biases arise from: (1) diagnostic mismatch of sampling altitudes between model and observations, (2) failure of model to produce sufficiently low ozone concentrations under rainy conditions, and (3) poor representation in the model of the diurnal cycle in boundary layer mixing and in stomatal conductance. Of these issues, (1) is the most straightforward to address, and the paper suggests an approach to diagnose more accurately the simulated ozone concentration at the altitude of the observations. This is a good recommendation that should be considered for adoption in other air quality modeling studies. The mechanisms responsible for issue (2) are not adequately addressed in the paper. The focus in the paper is on increased vertical stability resulting from evaporative cooling, but an alternative hypothesis of increased (non-stomatal) deposition of ozone under wet conditions is not adequately explored. Finally, the paper proposed to address issue (3) by focusing comparisons on afternoon ozone values rather than MDA8 in the short term, and by improving the representation of boundary layers (and presumably stomatal conductance) in the longer term.

This paper, with sufficient revisions, could provide a useful contribution to the literature and would help to address a long-standing bias of atmospheric chemistry models in simulating surface ozone. Specific comments and suggestions are included below.

1. Introduction

page 2, lines 6-7 – Mention that the use of MDA8 for comparisons between models and observations was intended to remove (some of) the known biases in the simulation of nighttime ozone, as opposed to comparing 24-hour averages.

We know of no reference that makes this point about the MDA8 metric, and would need to be provided such a reference before changing the text.

2. Comparing simulations of mixed layer and MDA8 surface ozone

p.3, l. 9 – Is this different from how local solar time is treated in the observations? It is not and we removed this text to avoid confusion.

p.3, l.14 – Add "sampled at lowest model grid level (zm=65m AGL)" here. Added.

3. Correcting for surface layer gradients

p.4, 1.5 – Not really a "correction." Instead, it is a transfer function from z=65m to z=10m. Changed correction to "transfer function" in this instance.

4. Segregating rainy conditions

p.4, l. 14-16 – But, this doesn't establish that transport from GoM to SE US is correct in model (e.g., nighttime low-level jet).

We clarify by adding the following text to p5, 1.7. "In addition, the occurrence of low values of observed MDA8 ozone is distributed across the CASTNET sites in the Southeast and is not related to distance from the Gulf."

p.4, l. 19-23 – How similar are the dates diagnosed as rainy/dry in the model vs obs? That is, how well does the model simulate daily variability of precipitation?

We added the following on p. 4, 123 - Rainy and dry days in the model are diagnosed from the GEOS-FP data, and do not necessarily coincide with rainy and dry days in the observations; our purpose here is to compare how rain affects ozone in the observations and in the model. 15% of observation days and 10% of model days are rainy."

p.4, 1.27 – Add "increased" before "vertical stratification." We revised the text to read on p. 5, 1.22. **"It may instead reflect increased atmospheric stability."**

p.4, 1.27-29 - See also Clifton et al. (2017), who say:

Recent field-based evidence suggests that nonstomatal processes include ... aqueous chemical reactions on vegetation and soil [Fowler et al., 2009; Ganzeveld et al., 2015; Fumagalli et al., 2016].

Thank you for these suggestions – we feel our current reference list for this point is appropriate and do not find that these additional references support our argument but did include a citation for Clifton et al, 2019 that does appear to be appropriate (p.5, line 25).

p.4, l. 27-29 and elsewhere – Need more description of GC dry deposition scheme. In particular, how does dry deposition velocity respond to moisture (incl. rainfall, soil moisture, dew on leaves, relative humidity, vapor pressure deficit)? Are there potentially missing processes that could increase ozone deposition velocities under wet conditions?

We added the following clarification on p3, 1.11 - **"Dry deposition of ozone follows a standard resistance-in series scheme (Wesely, 1989; Wang et al., 1998) where the surface resistance depends on leaf area and stomatal opening (itself dependent on temperature and solar radiation)."**

5. Accounting for diurnal bias p.5, 1.6 – Or other large NOx emission sources? We added "**or industrial sources.**"

p.5, l. 8-12 – Couldn't this also result from (excessive) mixing of ozone from throughout the first model grid level down to the surface. The rescaling to 10-m values wouldn't correct for this. Also, how valid are the assumption used in this rescaling under stable nighttime conditions?

We added the following clarification on p6, 1.5 -"(thus accounting for the vertical gradient within the lowest model level, including for stable conditions as given by equations (1)+(2)+(3c))."

p.5, l. 11-12 – Explain what drive the (diurnal) variations in stomatal conductance in GC. See above addition - p3, l.11 – "Dry deposition of ozone follows a standard resistance-in series scheme (Wesely, 1989; Wang et al., 1998) where the surface resistance depends on leaf area and stomatal opening (itself dependent on temperature and solar radiation)."

6. Implications p.6, 1.15 – Add "(e.g., non-stomatal dry deposition pathways)" here. Added.

p.6, 1.15-16 – Is the evening bias in models due exclusively to errors in vertical mixing, or could errors in the timing of the shutdown of stomatal conductance also play a role?

We added the following clarification on p7, line 20 – "and in the deposition of ozone to wet surfaces and at night."

p.6, l. 17-18 – Is better near-surface vertical resolution in models needed?

See response to above reviewer – added the following on p7, 1.20 - Finer model vertical resolution in the surface layer could improve the representation of the gradient but would require smaller model integration time steps.

p.6, 1.22 – Not discussing predictions elsewhere in paper. Change "predicted with confidence" to "simulated more accurately."

We removed this text and simplified the concluding sentence to the following: "Further model evaluation with MDA8 ozone for air quality applications should be contingent on proper representation of the modelozone diurnal cycle."

**Systematic bias in evaluating chemical transport models with maximum daily 8-hour average (MDA8) surface ozone for air quality applications: a case study with GEOS-Chem v9.02**

Katherine R. Travis1, Daniel J. Jacob2,3

5 1Department of Civil and Environmental Engineering, Massachusetts Institute of Technology, Cambridge, MA, USA 2School of Engineering and Applied Sciences, Harvard University, Cambridge, MA, USA 2Department of Earth and Planetary Sciences, Harvard University, Cambridge, MA, USA

Correspondence to: K. R. Travis (ktravis@mit.edu)

Abstract. Chemical transport models typicallyfrequently evaluate their simulation of surface ozone with observations of the maximum daily 8-hour average (MDA8) concentration, which is the standard air quality policy metric. This requires successful simulation of the surface ozone diurnal cycle including nighttime depletion, but models are generally biased high at night because of difficulty in resolving the stratified conditions near the surface. We quantify the problem with the GEOS Chem model foroften have difficulty simulating this diurnal cycle for a number of reasons including (1) vertical grid structure in the surface layer, (2) timing of changes in mixed layer dynamics and ozone deposition velocity across the day-night transition, (3)
 poor representation of nighttime stratification, (4) uncertainties in ozone nighttime deposition. We analyze the problem with

- the GEOS-Chem model, taking as representative case study the Southeast US during the NASA SEAC4RS aircraft campaign in August-September 2013. The model is unbiased relative to the daytime mixed layer aircraft observations but has a  $\pm$ 5mean  $\pm$ 8 ppb bias at its lowest level (65 m) relative to MDA8 surface ozone observations. The bias can be corrected to +5 ppb by implicit sampling of the model alsoat the 10 m altitude of the surface observations. The model does not capture frequent
- 20 observed occurrences of <20 ppb MDA8 surface ozone on rainy days-, possibly because of unaccounted enhancement of ozone deposition to wet surfaces. Restricting the surface ozone evaluation to dry days still shows inconsistencies with MDA8 ozone because of model errors in the ozone diurnal cycle. Restricting the evaluation to afternoon hours and dry days ozone completely removes the bias. Better understanding of surface We conclude that better representation of diurnal variations in mixed layer stratificationdynamics and ozone depletion under nighttime and rainy conditions deposition velocities is needed. Resolving the
- 25 timing of the day-night transition in atmospheric stability and its correlation with plant stomata closure is critical. in models to properly describe the diurnal cycle of ozone.

**1** Introduction**

Ground-level ozone is harmful to human health and vegetation. It is produced when volatile organic compounds (VOCs) and carbon monoxide (CO) are photochemically oxidized in the presence of nitrogen oxide radicals ( $NO_x \equiv NO+NO_2$ ). Ozone air

quality standards in different countries are generally formulated using the maximum daily 8-hour average concentration (MDA8) as a metric. In the US, the current ozone National Ambient Air Quality Standard (NAAQS) set by the Environmental Protection Agency (EPA) is 70 ppb as the fourth-highest MDA8 concentration per year, averaged over three years (EPA, 2015). Exceedances of the standard generally occur during daytime due to photochemical production and to entrainment of

5 elevated ozone from aloft (Kleinman, et al., 1994). Ozone is depleted at night due to deposition and chemical loss in a shallow surface layer capped by a stratified atmosphere.

Air quality agencies rely on chemical transport models (CTMs) to identify the most effective emission reduction strategies for ozone pollution. CTMs predict surface ozone concentrations on the basis of NOx, VOC, and CO emissions, accounting for

- 10 chemistry and meteorological conditions. CTMs tend to overestimate surface ozone, particularly in the Southeast United States (Fiore et al., 2009), and a variety; Makar et al., 2017). Some of reasons for this overestimate are examined is likely due to bias in the NOx 
[revised manuscript text omitted]

---

## Author Response (AR2)

Editor comment: One issue still remains before it can be published, though: neither a Dropbox nor a GitHub repository are acceptable solutions for archiving source code, input data and model output. Please read and comply with the GMD code and data policy (https://www.geoscientific-model-development.net/about/code_and_data_policy.html).

We thank the editor for his review of this paper, and to the above point have now uploaded the model code and output to zenodo and it now has a doi. We added the following to the Data Availability: **
[revised manuscript text omitted]